# Effect of High Fat Diet on Disease Development of Polycystic Ovary Syndrome and Lifestyle Intervention Strategies

**DOI:** 10.3390/nu15092230

**Published:** 2023-05-08

**Authors:** Yingxue Han, Hao Wu, Siyuan Sun, Rong Zhao, Yifan Deng, Shenming Zeng, Juan Chen

**Affiliations:** 1National Engineering Laboratory for Animal Breeding, Key Laboratory of Animal Genetics and Breeding of the Ministry of Agriculture, College of Animal Science and Technology, China Agricultural University, Beijing 100193, China; 2Key Laboratory of Precision Nutrition and Food Quality, Department of Nutrition and Health, China Agricultural University, Beijing 100190, China

**Keywords:** high-fat diet, PCOS, lifestyle, diet strategies, hypothalamic–pituitary–gonadal axis, hyperinsulinemia

## Abstract

Polycystic ovary syndrome (PCOS) is a prevalent endocrine and metabolic disorder that affects premenopausal women. The etiology of PCOS is multifaceted, involving various genetic and epigenetic factors, hypothalamic–pituitary–ovarian dysfunction, androgen excess, insulin resistance, and adipose-related mechanisms. High-fat diets (HFDs) has been linked to the development of metabolic disorders and weight gain, exacerbating obesity and impairing the function of the hypothalamic–pituitary–ovarian axis. This results in increased insulin resistance, hyperinsulinemia, and the release of inflammatory adipokines, leading to heightened fat synthesis and reduced fat breakdown, thereby worsening the metabolic and reproductive consequences of PCOS. Effective management of PCOS requires lifestyle interventions such as dietary modifications, weight loss, physical activity, and psychological well-being, as well as medical or surgical interventions in some cases. This article systematically examines the pathological basis of PCOS and the influence of HFDs on its development, with the aim of raising awareness of the connection between diet and reproductive health, providing a robust approach to lifestyle interventions, and serving as a reference for the development of targeted drug treatments.

## 1. Introduction

In human diets, the definition of a high-fat diet (HFD) is the intake of fat calories accounting for 30% to 75% of the total caloric intake [1,2,3,4,5,6]. A HFD is a type of diet that is characterized by an excessive intake of fat, particularly saturated fat. Consuming too much saturated fat can increase the levels of bad cholesterol in the body, which can lead to an increased risk of developing heart disease and other health problems. In addition, HFDs are often associated with an excessive intake of calories, which can lead to weight gain and obesity. It is recommended that fat intake should be in the proportions between 20% and 35% of total calories [7]. HFDs, especially when coupled with nutrient excess, can trigger lipid accumulation in both fat and non-fat tissues [8]. Once the lipid storage capacity of non-fat tissues reaches its limit, lipotoxicity can set in, leading to cellular dysfunction, cell death, and the development of obesity and its related diseases [9].

PCOS is a prevalent endocrine disorder among women at reproductive age, which can contribute to menstrual irregularities or an absence of menstruation, high androgen levels, and related metabolic and psychological consequences. PCOS is frequently associated with abdominal adiposity, insulin resistance, obesity, metabolic dysfunction, and cardiovascular risk factors. Three definitions of PCOS are currently in use, with the Rotterdam criteria being the most widely accepted and endorsed by numerous scientific societies and health authorities [10,11,12]. According to the Rotterdam criteria, a diagnosis of PCOS requires the presence of at least two of the following criteria in women: clinical and/or biochemical hyperandrogenism, ovulatory dysfunction, and polycystic ovarian morphology (PCOM). The Androgen Excess and PCOS Society (AE-PCOS) criteria, introduced in 2006, provides an alternative approach to the diagnosis of PCOS. The AE-PCOS criteria require the presence of hyperandrogenism, along with either oligo-ovulation or polycystic ovaries. The diagnosis of PCOS requires the exclusion of other endocrine disorders, such as thyroid or adrenal disorders, that may cause similar symptoms. Clinicians should also consider the potential of confounding factors, such as medications or weight changes, that may affect the presentation and interpretation of PCOS diagnostic criteria [11,13]. Women with PCOS are more likely to have problems with insulin resistance, which can lead to high levels of insulin in the blood and other metabolic issues such as high cholesterol, type 2 diabetes, and obesity. In addition to metabolic health risks, women with PCOS are also at higher risk of developing psychological and emotional problems, such as depression and anxiety. This may be due to the chronic stress associated with the condition, as well as the hormonal imbalances and social stigma. The psychological impact of PCOS can further exacerbate the negative effects on quality of life and well-being [14].

HFDs and the resulting obesity are neither necessary nor sufficient conditions for the PCOS phenotype [15]. Women with PCOS often have hyperinsulinemia, abdominal fat accumulation, and metabolic disorders such as obesity. Moreover, overweight or obese PCOS patients, particularly those with excessive androgens [16], may have more severe metabolic reproductive outcomes [17].

## 2. Materials and Methods

### Bibliographic Search Methods

A systematic literature search was conducted to identify relevant studies addressing the relationship between HFDs and PCOS, as well as potential dietary and lifestyle interventions for PCOS. The search was performed using the PubMed database, a comprehensive biomedical literature database maintained by the National Library of Medicine.

To identify relevant studies, a combination of medical subject headings (MeSH terms) and free-text keywords were used. The search strategy employed the following search terms: “high-fat diet”, “PCOS”, “pathology of PCOS”, “dietary strategy and PCOS”, and “lifestyle and PCOS”. These keywords were selected based on their relevance to the research question and the aim of the literature review.

The search was limited to articles published in the English language and peer-reviewed articles from 1989 to 2023. The inclusion criteria for selecting relevant articles were based on the following criteria: (1) studies that investigated the relationship between HFDs and PCOS or the potential dietary and lifestyle interventions for PCOS, (2) studies that reported original research findings, and (3) studies that were published in peer-reviewed journals. The initial search yielded a total of 829 articles. Duplicate articles were removed, and the remaining articles were screened based on their titles and abstracts for relevance. The full text of the relevant articles was then reviewed for inclusion in the literature review. In addition to the database search, a manual search of the reference lists of the included articles was performed to identify any additional studies that met the inclusion criteria.

Overall, the literature search strategy employed in this review was designed to identify relevant articles that provide insights into the underlying mechanisms linking HFDs to PCOS, as well as the potential dietary and lifestyle interventions that may alleviate PCOS symptoms and metabolic dysfunctions.

## 3. Pathophysiology of PCOS

### 3.1. Androgen Excess and Insulin Resistance

PCOS is essentially identified by a cluster of symptoms and physical indications resulting from elevated androgen levels and impaired ovarian function, which lacks any specific diagnosis [10]. Hyperandrogenemia is regarded as a significant factor that exacerbates reproductive symptoms and fosters the development of metabolic syndrome in PCOS. Excessive androgen production mainly originates from the ovaries and adrenal glands. Research has reported that excessive androgen production from the ovaries is one of the major inducing factors in PCOS [18,19,20,21,22,23]. Females with PCOS exhibit heightened frequencies of gonadotropin-releasing hormone (GnRH) pulses, culminating in excessive luteinizing hormone (LH) secretion, which in turn provokes androgen production by the follicular cells of the ovaries. Follicle-stimulating hormone (FSH), which remains unaltered (or low), inhibits follicle enlargement and maturation, resulting in follicular stasis and the formation of polycystic ovaries (PCOSM) along with infrequent or absent ovulation. A considerable number of preantral and small antral follicles enhance the production of anti-Müllerian hormone (AMH). An elevated level of AMH also amplifies GnRH neuronal activity and directly stimulates the secretion of LH, which potentially impairs the production of excessive ovarian androgens [13]. Mounting evidence indicates that excessive androgen production is not solely responsible for oligo-ovulation and cutaneous manifestations in this syndrome but also promotes insulin resistance and metabolic dysfunction in PCOS women through favorable effects on abdominal and visceral obesity [24,25,26].

Hyperinsulinemia caused by insulin resistance is also inherent in PCOS [27]. About 1–2/3 of PCOS patients exhibit different degrees of insulin resistance (although insulin resistance itself is not a decisive feature of PCOS) [27,28,29,30,31]. Hyperinsulinemia caused by insulin resistance acts on theca cells, stimulates androgen production, and exacerbates hyperandrogenemia. Meanwhile, insulin and androgens synergistically induce premature luteinization of granulosa cells and stimulate lipid synthesis [14]. Hyperandrogenemia causes excessive LH production, which in turn stimulates theca and luteinized granulosa cells, exacerbating preexisting hyperandrogenemia [13,14]. Moreover, LH prompts luteinized granulosa cells to secrete estradiol and suppresses FSH secretion [32]. These alterations in granulosa cell function, potentiated by hyperinsulinemia, are implicated in the manifestation of polycystic ovary morphology (PCOM) and hindered ovulation [14]. To summarize, the incidence of PCOS increases in any disease characterized by systemic hyperinsulinemia [10].

### 3.2. Heterogeneity of PCOS

PCOS is a complex and heterogeneous disorder with a multifactorial etiology that involves both genetic and environmental factors [10]. Numerous genetic variants have been associated with PCOS, including those that affect insulin signaling, hormone receptors, and steroidogenesis. However, the genetic factors alone cannot explain the high prevalence of the disorder, suggesting that environmental factors also play a role. Environmental factors that have been implicated in the development of PCOS include prenatal exposure to androgens, obesity, insulin resistance, and chronic inflammation. The precise mechanisms via which these factors contribute to the development of PCOS are not yet fully understood [33]. Additionally, PCOS is the result of interactions between primary androgen excess (manifested as excess androgen synthesis) and other factors such as obesity, abdominal adiposity, and insulin resistance [10]. In some cases, high androgen abnormalities alone can be severe enough to cause PCOS, even in lean women without visceral adiposity or insulin resistance (specific proportions provided). In other cases, the phenotype of PCOS only appears in the presence of triggering factors such as obesity, abdominal adiposity, insulin resistance, and/or hyperinsulinemia [34,35,36]. The severity of androgen secretion abnormalities can vary between individuals, resulting in a spectrum of PCOS symptoms. Therefore, the heterogeneity of PCOS symptoms can be attributed to differences in the extent of contributions from obesity and insulin resistance.

## 4. Effect of High-Fat Diet on Disease Development of Polycystic Ovary Syndrome

### 4.1. The Impact of High-Fat Diet on the Neuroendocrine Mechanisms of Female Reproduction

HFDs, irrespective of whether or not they leads to obesity, have been shown to compromise the performance and reproductive capacity of the hypothalamic–pituitary–ovarian (HPO) axis in women [9]. One study demonstrated that mice with diet-induced obesity (DIO) experienced a 60% decline in their spontaneous conception rates. However, this impairment could be rescued with the administration of exogenous gonadotropins, suggesting the involvement of a central mechanism [19]. The physiological activities of three organs, the hypothalamus, pituitary gland, and ovaries, regulate follicle maturation and its ovulation mechanism. Hormonal regulation in the hypothalamic–pituitary–ovarian system is governed by a negative-feedback axis. GnRH synthesized in the hypothalamus is released into the portal circulation of the pituitary gland, thereby promoting gonadotropin release. These hormones are secreted in a pulsatile rhythm, with low-frequency gonadotropin pulses causing FSH secretion, and high-frequency pulses leading to the secretion of LH by the anterior pituitary [37]. In many PCOS patients, the fundamental pathological and physiological changes observed are excessive androgen production by the ovaries, primarily caused by excessive secretion of LH, while FSH secretion is normal or slightly lower, leading to an increased LH/FSH ratio. LH acts directly on the follicular membrane cells of the ovary, increasing the function of intracellular branched-chain cleavage P450c17a and causing the follicular membrane cells to produce excessive androgens [13].

The adipose tissue is acknowledged as an endocrine gland that generates a range of bioactive substances, which are commonly referred to as adipokines. These substances include but are not limited to adiponectin, visfatin, leptin and resistin [38]. Overproduction of proinflammatory adipokines may result in persistent low-grade metabolic inflammation, which has the potential to impair the proper functioning of the HPO axis [39]. Specifically, elevated adipokines can overstimulate kisspeptin neurons, resulting in impaired GnRH release frequency. At the pituitary level, the abnormal release of gonadotropins reflects a local increase in androgens, which plays a role in the development of symptoms associated with PCOS. Comparative studies on female rats exposed to HFDs have shown that HFDs can modify ovarian cycle activity by increasing LH pulse frequency and enhancing kisspeptin expression [40,41].

HFDs can induce elevated insulin levels, which may upregulate the GnRH pulse amplitude of the hypothalamus, thereby increasing the secretion of LH from the pituitary gland. Several studies have reported that animals fed a HFD exhibit hyperinsulinemia, which can stimulate androgen biosynthesis and release by enhancing the effectiveness and pulse amplitude of LH. Additionally, the significantly increased frequency of LH pulses resulting from high insulin levels may alter the normal dynamics of follicular development, thereby delaying or inhibiting the selection of dominant follicles while increasing the recruitment of smaller follicles [42]. While obesity does not determine the development of PCOS, it may contribute to exacerbating both metabolic and reproductive outcomes associated with PCOS, particularly when it is centered around the abdominal region (known as visceral obesity) [15].

### 4.2. High-Fat Diet and Metabolic Imbalance

Sustained HFD intake is linked with the emergence of obesity and metabolic disturbances, such as hyperglycemia, hypertension, insulin resistance, dyslipidemia, and endocrine dysfunction. Obesity is a prevalent and pressing public health issue that affects individuals in childhood and adulthood globally [43,44]. According to the World Health Organization, the occurrence of overweight and obesity among females aged 5–19 years has considerably risen over the years. Specifically, the rate has risen from 4% in 1975 to 18% in 2016 [45]. This alarming trend highlights the need for urgent intervention and prevention strategies to address the health consequences associated with obesity. Notably, obesity is commonly categorized using body mass index (BMI), but BMI may not provide an accurate assessment of body composition or fat distribution [46]. Measuring waist circumference (WC) and calculating the WC/height ratio (WC/H) can provide valuable information on abdominal obesity, which is not accurately assessed by BMI. A WC/H ratio greater than 0.5 is considered indicative of visceral obesity, which is more likely to be present in patients with PCOS. Even in lean women with PCOS, higher levels of visceral adipose tissue (VAT) have been observed, which could be linked to poorer metabolic outcomes. Therefore, measuring WC and calculating the WC/H ratio may be useful in assessing abdominal obesity and predicting metabolic risks in patients with PCOS [47,48]. A plethora of animal studies investigating the reproductive effects of HFD have consistently reported increased serum insulin levels [49,50,51,52], elevated blood glucose levels [53,54,55], and impaired insulin sensitivity [56,57,58].

The deleterious impact of HFDs results primarily from thier effect on the insulin signaling pathway [59,60,61,62]. Research has demonstrated that insulin plays a role in promoting gonadotropin production in the ovary [63,64]. Insulin receptors are expressed in the granulosa and theca cells, and insulin has been shown to influence ovarian steroidogenesis [64,65]. Specifically, insulin can induce androgen production in theca cells and promote the proliferation of GCs, leading to increased estradiol production [66]. Overstimulation of the insulin pathway by HFDs enhances LH activity on theca cells, leading to excessive androgen production and subsequent granulosa cell differentiation disorder in female mice [62,67,68,69], rats [70,71,72], pigs [60], humans [73], and non-human primates [74,75,76], which affects the follicular development process. Specifically, insulin signaling appears to stimulate the phosphatidylinositol 3-kinase (PI3K) pathway in both the ovary and pituitary gland, rather than the extracellular signal-regulated kinase (ERK) pathway [57]. Although other factors in the insulin signaling cascade may also contribute, Kit ligand (KITLG) is known to be an activator of the PI3K pathway [54]. Evidence suggests that both KITLG and insulin can activate PI3K signaling in the context of obesity [77]. Additionally, after HFD exposure, leptin levels increase and activate PI3K signaling [42,78].

Prolonged consumption of a HFD may lead to the accumulation of abdominal adipose tissue, thereby promoting obesity development. It is known that obesity induces insulin resistance and stimulates the production of testosterone from circulating androgens, while suppressing gonadotropin secretion [15]. Testosterone, generated in such conditions, has been shown to stimulate the buildup of visceral fat in women with PCOS by inhibiting fat breakdown and promoting fat synthesis [14]. This process ultimately leads to insulin resistance and metabolic dysfunction in affected individuals, as demonstrated by numerous studies [15,79,80]. Enlargement of adipocytes within the abdominal region in obesity has been demonstrated to enhance the release of pro-inflammatory cytokines from adipose tissue monocytes in response to glucose and saturated fat intake [81]. In the context of PCOS, this phenomenon is further exacerbated by hyperandrogenism [82]. Additionally, obesity-related compensatory hyperinsulinemia has been demonstrated to sensitize ovarian follicular cells to stimulate LH, which, in conjunction with the impact of excess insulin on promoting fat synthesis and inhibiting fat breakdown, ultimately results in the development of obesity in individuals with PCOS [14].

The research indicates that PCOS women who are overweight or obese experience more severe symptoms compared to those with a normal weight [15]. Specifically, these women exhibit an increased free androgen index (FAI), higher insulin resistance, elevated fasting blood glucose levels, lower levels of sex hormone-binding globulin (SHBG), higher levels of testosterone (T), and more prominent hirsutism [83,84].

### 4.3. Effect of High-Fat Diet on Ovarian Function

One key feature of PCOS is anovulation, which is the inability to regularly ovulate. Research studies have demonstrated that HFDs can adversely affect follicular development and ovulation in females [9]. Asemota et al. found that mice fed a HFD had a reduced number of oocytes following superovulation [85]. Additionally, HFDs have been linked to alterations in the hormonal environment of the ovary, including reduced FSH concentrations and elevated androgen levels, which can lead to impaired follicle development and decreased oocyte quality [9]. One study found that a HFD led to a decreased ovarian reserve and reduced oocyte quality in mice [86]. Similarly, a review mentioned that in studies conducted on overweight and obese women with PCOS, HFDs were linked with increased insulin resistance and reduced follicular responsiveness to FSH, leading to impaired follicle development [33]. In another study of non-human primates, a HFD was found to result in a decreased number of developing follicles, as well as a decreased proportion of follicles that had reached the antral stage, which are necessary for ovulation [87]. Overall, these findings suggest that HFDs can have negative effects on the quantity and quality of developing ovarian follicles, potentially leading to impaired fertility [8,56].

In addition, studies have shown that HFDs can also impair the ovulatory function of the ovaries, based on the following points. Firstly, HFDs have been found to decrease the expression of genes associated with normal ovulation function, which can lead to reproductive failure [88]. Secondly, HFDs can disrupt the recruitment of immune cells to the ovary that are essential for ovulation. Specifically, the molecular factor MCP-1, which is involved in monocyte recruitment from the blood into the ovary, is dysregulated under HFD conditions, leading to impaired ovulatory function [89]. Finally, proper lipid metabolism is crucial for generating sufficient energy to support oocyte maturation and ovulation, and HFDs can impair this process. The results of these research studies indicate that HFD intake may have adverse effects on female reproductive success and ovulatory function.

An overall summary of the information described above is reported in Figure 1.

## 5. The Role of Lifestyle Modification in PCOS

Lifestyle modification is a critical intervention for PCOS women who are overweight or obese that can result in weight loss, reduced insulin resistance and circulating androgens, and improved menstrual and hirsutism symptoms [15]. Initial interventions for PCOS include lifestyle modification with dietary changes, exercise, and behavioral programs; however, with the exception of bariatric surgery, few treatments lead to permanent weight loss. Despite the fact that a weight loss of 5–10% may improve symptoms of PCOS, severely obese individuals may require a weight loss of 25–50% [90,91].

### 5.1. Dietary Strategies

It is recommended that women with PCOS make dietary modifications that can improve their metabolic health and reduce symptoms associated with the condition. Recommendations for women with PCOS are based on previous research. A balanced diet that is low in saturated and trans fats, and high in unsaturated fats can help achieve these goals. [92]. Additionally, increasing the intake of fiber-rich foods such as fruits, vegetables, whole grains, and legumes can help regulate insulin levels and promote weight loss. Consuming lean protein sources such as fish, poultry, and plant-based proteins is also recommended. Finally, limiting the intake of simple sugars, processed foods, and salt can help improve insulin sensitivity, lower blood pressure, and reduce the risk of cardiovascular disease. Overall, a well-balanced diet that includes a variety of nutrient-rich foods can help women with PCOS manage their symptoms and improve their overall health [93].

Various dietary interventions have been suggested to improve metabolic health and alleviate PCOS symptoms. Among these interventions, low-glycemic-index (GI) diets have been shown to improve insulin sensitivity and menstrual regularity in women with PCOS [94,95]. A study by Marsh et al. [96] compared changes in insulin sensitivity and clinical outcomes after weight loss in 96 PCOS women who consumed either a low-GI diet or a traditional healthy diet. The low-GI diet group showed significant improvements in insulin sensitivity and menstrual regularity compared to the traditional healthy diet group. Low-calorie diets have also been associated with weight loss and improved insulin sensitivity in women with PCOS. While dietary interventions can improve metabolic health in women with PCOS, further research is needed to establish the optimal dietary approach for PCOS management. Individualized dietary counseling based on the patient’s metabolic profile and personal preferences may be beneficial [45]. The impact of different types of diets on the development of PCOS is presented in Table 1.

Moreover, the composition of the diet, specifically the quantity and quality of carbohydrates consumed, can impact insulin sensitivity, independently of weight loss [45]. A diet abundant in fiber and complex carbohydrates, particularly those from unrefined sources, has been correlated with enhanced insulin sensitivity [115]. Research has shown that consuming soluble dietary fiber, in particular, has been observed to lead to delayed gastric emptying, impaired nutrient assimilation, including glucose assimilation, and enhanced feelings of fullness [116]. Plant-based foods, which are rich in dietary fiber and promote glycemic control, are also abundant in phytochemicals, such as polyphenols, that can reduce hyperglycemia, and enhance both acute insulin response and insulin sensitivity [117].

In addition, the quality of dietary fat affects insulin sensitivity and related metabolic abnormalities. Epidemiological evidence and intervention studies have clearly demonstrated that saturated fat worsens insulin resistance, whereas mono- and poly-unsaturated fatty acids improve insulin resistance by altering cell membrane composition [118]. A recent multicenter study (KANWU) showed that replacing a diet high in saturated fatty acids with one high in mono-unsaturated fatty acids ameliorates insulin sensitivity in healthy individuals, whereas modest supplementation with omega-3 fatty acids does not have a significant impact on insulin sensitivity [118]. It is widely acknowledged that not all animal-derived fats have equivalent effects. Omega-3 fatty acids, specifically eicosapentaenoic acid (EPA) and docosahexaenoic acid (DHA), are found to be associated with improved insulin resistance. Foods high in omega-3 fatty acids are typically fatty fish, such as salmon, tuna, and sardines. However, alpha-linolenic acid, the precursor of EPA and DHA, is present in some plant-based foods such as nuts and seeds. In women with PCOS, dietary supplementation with α-lipoic acid, N-acetylcysteine, and omega-3 fatty acids has been found to exhibit antioxidant and anti-inflammatory properties and enhance insulin sensitivity [119].

Regardless of which specific diet plan is used, it should adhere to the basic principles of a healthy diet that promotes physiological metabolic balance and supports overall health and recovery from illnesses.

### 5.2. Physical Activity

A meta-analysis of lifestyle adjustments for PCOS indicates their beneficial effects on body composition, high androgen levels, and insulin resistance in women with PCOS [120]. The primary benefits of exercise include weight loss, improved glucose metabolism, and reduced insulin resistance. Indirect benefits include lower levels and actions of androgens, improved endocrine function, and restored fertility. In addition, organized exercise, based on dietary calorie restriction, can improve ovulation [121] and glucose tolerance [122], restore ovulation [123], reduce weight [124], and decrease skinfold thickness [125].

### 5.3. Mental Health

Psychological and social factors are crucial in disease treatment. Many studies have demonstrated that patients with PCOS are affected by various aspects of mental and psychological health, including psychological disorders (such as depression and anxiety), eating disorders, self-esteem, body dissatisfaction, and appearance. Recent investigations found that the incidence of depression in PCOS patients ranged from 14% to 67%, and that the likelihood of depression symptoms was four times higher than that of age-matched women in the control group [126]. Domestic and international data indicate that some PCOS patients experience psychological disorders, such as depression or anxiety, to various degrees. These may be associated with the abnormal expression of inflammatory factors caused by the “metabolic inflammation” status or immune abnormalities of PCOS patients. In PCOS patients with comorbid psychological disorders, the interaction between pathophysiological and pathological psychological factors leads to a low quality of life, the exacerbation of reproductive functional disorders (such as menstrual irregularity, amenorrhea, and ovulatory dysfunction and, metabolic disorders, and other physical symptoms. Such patients experience endocrine and immune stress over an extended period [71]. Therefore, PCOS patients should maintain an optimistic mood, avoid depression and anxiety, relax themselves, reduce stress, maintain confidence in life, and therefore maintain a healthy psychological state. Additionally, patients with significant symptoms should receive specific and timely psychological counseling and medication treatment from a psychiatric department [118].

## 6. Conclusions and Perspective

Given the prevalence of HFDs and the increasing incidence of obesity among women of reproductive age, it is crucial to understand the effects of HFDs on the development and progression of PCOS. This review highlights the need to unravel the complex mechanisms of PCOS pathology and the interactions among the related contributing factors in order to recognize the deleterious impact of HFDs on PCOS. Furthermore, it demonstrates how HFDs can exacerbate all metabolic and reproductive outcomes of PCOS patients. HFDs can cause insulin resistance and compensatory hyperinsulinemia, leading to increased fat synthesis and decreased fat breakdown. Additionally, HFDs increase the production of inflammatory lipid factors, further exacerbating fat synthesis. According to the aforementioned, there is a potential correlation between HFDs and PCOS; however, the exact mechanisms and impacts require further investigation. In the future, it is imperative to continue in-depth explorations of the association between HFDs and PCOS, and incorporate these into a comprehensive management approach for PCOS, encompassing multidisciplinary interventions such as dietary interventions, physical exercise, weight loss, and medication therapy, to achieve optimal clinical outcomes. Additionally, enhancing public health awareness and promoting healthy dietary habits can aid in the prevention of PCOS occurrence and progression. Future research should also focus on the effects of different types and proportions of fats on PCOS to develop evidence-based and practical dietary intervention strategies for more effective prevention and management of PCOS. In conclusion, further rigorous research and application of the relationship between HFDs and PCOS in clinical practice are warranted to facilitate the health management and prevention of PCOS among affected individuals.

## Figures and Tables

**Figure 1 nutrients-15-02230-f001:**
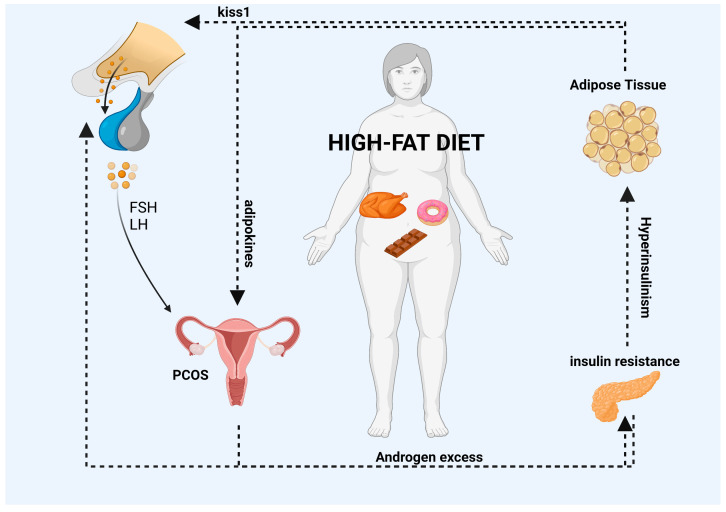
HFDs can lead to the accumulation of adipose tissue and an increase in the systemic release of adipokines. The interaction between PCOS and abdominal obesity may result in a vicious cycle (as indicated by the black arrows in the diagram) in which excess androgens promote visceral adiposity. This, in turn, may promote excess androgens of ovarian and/or adrenal origin via various mediators, including autocrine, paracrine, and endocrine pathways, or indirectly through the induction of insulin resistance and hyperinsulinemia. Moreover, elevated levels of adipokines can impair the GnRH pulse by overstimulating kisspeptin neurons, which can have an impact on the release of FSH and LH gonadotropins from the pituitary gland and the secretion of E2 and P4 hormones from the ovaries. Finally, high levels of insulin, which are a characteristic feature of HFDs, can increase the amplitude of the GnRH pulse in the hypothalamus, leading to an increase in LH secretion from the pituitary gland. List of abbreviations included in the figure: follicle stimulating hormone (FSH); luteinizing hormone (LH); polycystic ovary syndrome (PCOS). Image created with BioRender.com.

**Table 1 nutrients-15-02230-t001:** Effects of different dietary types on the development of PCOS.

Dietary Type	Nutrient Distribution	Influence	Ref.
High-GI ^1^	High in carbohydrate, low in protein, and low in fat	Increases insulin resistance, worsens blood sugar control, and increases risk of cardiovascular disease	[97,98,99]
Low-GI	Low in carbohydrates, high in protein, low in fat	Improves insulin sensitivity, improves blood sugar control, aids in weight loss	[100,101,102]
Mediterranean-style	Low in saturated fatty acids, high in monounsaturated fatty acids, high in fiber; low-GI foods	Improves insulin sensitivity, reduces risk of cardiovascular disease	[103,104]
High-protein	High in protein, low in carbohydrates, low in fat	Improves insulin sensitivity, aids in weight loss	[94,105]
Low-calorie	Low total energy intake	Improves blood sugar control, promotes weight loss	[106,107,108]
Saturated fatty acids	High in saturated fatty acids	Increases risk of cardiovascular disease	[109,110,111]
High-fiber	High in fiber; low-GI foods	Improves metabolic control, reduces risk of cardiovascular disease	[112,113,114]

^1^ glycemic index, GI.

## Data Availability

No new data were created or analyzed in this study. Data sharing is not applicable to this article.

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
