# Peer review of "Effect of High Fat Diet on Disease Development of Polycystic Ovary Syndrome and Lifestyle Intervention Strategies"

_nutrients, 2023, doi:10.3390/nu15092230_

Round 1

Reviewer 1 Report

The authors propose an interesting review paper about PCOS, HFD and lifestyle intervention strategies. 

1-) About the literature search: Did the search strategy includes which keywords?   Could you provide an idea of the search strategy adopted? Please, include in the manuscript.

2-) Please standardize the acronyms throughout the text. Some places say high-fat diet and others HFD. Check the other acronyms.

3-) Minor revision is required. 

Minor revision is required before publication. 

Author Response

Point 1: About the literature search: Did the search strategy includes which keywords? Could you provide an idea of the search strategy adopted? Please, include in the manuscript.

Response 1: Regarding your first comment on the literature search, we apologize for not including our search strategy in the manuscript. We have now revised the manuscript to include a detailed description of the keywords and search strategy used in our literature search and we have highlighted the changes in the text in yellow.

The following is an introduction to the literature search methods:

“A systematic literature search was conducted to identify relevant studies ad-dressing the relationship between HFD and PCOS, as well as potential dietary and lifestyle interventions for PCOS. The search was performed using the PubMed data-base, a comprehensive biomedical literature database maintained by the National Li-brary of Medicine.

To identify relevant studies, a combination of medical subject headings (MeSH terms) and free-text keywords were used. The search strategy employed the following search terms: "high-fat diet," "PCOS," "pathology of PCOS," "dietary strategy and PCOS," and "lifestyle and PCOS." These keywords were selected based on their rele-vance to the research question and the aim of the literature review.

The search was limited to articles published in English language and peer-reviewed articles from 1989 to 2023. The inclusion criteria for selecting relevant articles were based on the following criteria: (1) studies that investigated the relation-ship between high-fat diet and PCOS or the potential dietary and lifestyle interven-tions for PCOS, (2) studies that reported original research findings, and (3) studies that were published in peer-reviewed journals.The initial search yielded a total of 829 arti-cles. Duplicate articles were removed, and the remaining articles were screened based on their titles and abstracts for relevance. The full text of the relevant articles was then reviewed for inclusion in the literature review.In addition to the database search, a manual search of the reference lists of the included articles was performed to identify any additional studies that met the inclusion criteria.

Overall, the literature search strategy employed in this review was designed to identify relevant articles that provide insights into the underlying mechanisms linking high-fat diet to PCOS, as well as the potential dietary and lifestyle interventions that may alleviate PCOS symptoms and metabolic dysfunctions.”

Point 2: Please standardize the acronyms throughout the text. Some places say high-fat diet and others HFD. Check the other acronyms.

Response 2: Regarding your second comment on standardizing the acronyms, we have thoroughly checked the manuscript and made necessary changes to ensure that all acronyms are used consistently throughout the text. We have replaced "high-fat diet" with "HFD" wherever applicable, and also checked for other acronyms used in the text to ensure standardization.

Point 3: Minor revision is required.

Response 3: Thank you for pointing out the need for minor revisions. We have made the necessary changes and believe the manuscript is now improved.

Once again, we thank you for your insightful comments, and we hope that our revisions have addressed your concerns. Please let us know if there are any further issues or concerns that we can address.

Reviewer 2 Report

The authors present an interesting review on the impact of diet and lifestyle interventions for women suffering from PCOS. 

Some suggestions for revisions:

- the authors refer to high insulinemia throughout the manuscript, the appropriate terminology is hyperinsulinemia 

- please use patient first language throughout the manuscript, for example, women with overweight or obesity versus obese women

- the authors highlight the negative impacts of a HFD, however, there are studies that have found increased dietary fat intake (MUFA and PUFA) can improve fertility rates. The authors discuss briefly in 4.1 but it would be best if this distinction of a HFD is referring to a diet that is excessive in calories and saturated fats is made in the introduction. 

- 4.2. instead of Sports, Physical Activity would be a more relevant title

Minor English editing required in the Introduction section. 

Author Response

Point 1: the authors refer to high insulinemia throughout the manuscript, the appropriate terminology is hyperinsulinemia.

Response 1: Thank you for taking the time to review our manuscript and providing us with your insightful comments. We appreciate your valuable feedback, and we have carefully considered all of your suggestions and we have highlighted the changes in the text in green.

We have made the necessary revisions to use the appropriate terminology of hyperinsulinemia throughout the manuscript. Thank you for bringing this to our attention.

Point 2: please use patient first language throughout the manuscript, for example, women with overweight or obesity versus obese women.

Response 2: We appreciate your comment about using patient-first language throughout the manuscript. We have made the necessary changes to ensure that our language is respectful and inclusive, as suggested.

Point 3: the authors highlight the negative impacts of a HFD, however, there are studies that have found increased dietary fat intake (MUFA and PUFA) can improve fertility rates. The authors discuss briefly in 4.1 but it would be best if this distinction of a HFD is referring to a diet that is excessive in calories and saturated fats is made in the introduction.

Response 3: We appreciate your comment about the potential positive effects of monounsaturated and polyunsaturated fats on fertility rates. In response, we have revised the introduction to clarify that our focus is on diets that are excessive in calories and saturated fats, which have been associated with negative impacts on fertility.

Here's what's added:” HFD is a type of diet that is characterized by an excessive intake of fat, particularly saturated fat. Consuming too much saturated fat can increase the levels of bad choles-terol in the body, which can lead to an increased risk of developing heart disease and other health problems. In addition, a high-fat diet is often associated with an excessive intake of calories, which can lead to weight gain and obesity.”

Point 4: instead of Sports, Physical Activity would be a more relevant title.

Response 4: we agree that the title of section 4.2 should be revised to better reflect its content. We have revised the title to "Physical Activity" to better convey the section's focus.
